# Social Media Sentiments on Suicides at the New York City Landmark, *Vessel*: A Twitter Study

**DOI:** 10.3390/ijerph191811694

**Published:** 2022-09-16

**Authors:** Paul Yip, Yunyu Xiao, Yucan Xu, Evangeline Chan, Florence Cheung, Christian S. Chan, Jane Pirkis

**Affiliations:** 1The HKJC Centre for Suicide Research and Prevention, The University of Hong Kong, Hong Kong, China; 2Department of Population Health Sciences, Weill Cornell Medicine, NewYork-Presbyterian, New York, NY 10065, USA; 3Department of Psychology, The University of Hong Kong, Hong Kong, China; 4Centre for Mental Health, Melbourne School of Population and Global Health, Carlton VIC 3053, Australia

**Keywords:** suicide prevention, public sentiment, restriction of means, Twitter, social media

## Abstract

*Vessel* is a landmark created by Heatherwick Studio where visitors can enjoy views of New York City from different heights and perspectives. However, between February 2020 and July 2021, four individuals jumped to their deaths from the landmark. Effective preventive solutions have yet to be identified, and the site is currently closed. In this study, we examined the trajectory of public sentiment on the suicide-related activity at *Vessel* on Twitter by investigating the engagement patterns and identifying themes about the four suicides from February 2020 to August 2021 (*n* = 3058 tweets). The results show increased levels of discussion about each successive suicide case in the first 14 days following each incident (from 6 daily tweets for the first case to 104 for the fourth case). It also took longer for relevant discussions to dissipate (4 days for the first and 14 days for the fourth case, KS statistic = 0.71, *p* < 0.001). Thematic analysis shows a shift from expressions of emotion to urging suicide prevention actions in the third and fourth cases; additionally, we detected growing support for restricting means. We suggest that, prior to the reopening of *Vessel*, collective efforts should be made to install safety protections and reduce further suicide risks.

## 1. Introduction

On 15 March 2019, *Vessel*, a 16-story sculpture built as a visitor attraction in the Hudson Yards Redevelopment Project, was opened in New York City (see Figure 1) [1]. It comprises 154 intricately interconnecting flights of stairs, offering remarkable views of the city and the Hudson River from different heights, angles, and vantage points [2]. Unfortunately, aside from being a popular tourist attraction, it also gained attention because of the tragic incidents that took place on the site. Several youths (aged 14–24) have jumped from the structure, resulting in four suicides in the mere 29 months since it opened (see Figure 2) [3]. Although strategies including adding safety nets and raising the height of the glass barriers were advised by suicide prevention experts following the third death, no such netting and higher barriers have been installed [4]. Instead, minor actionable suicide prevention strategies were implemented, such as installing mental wellness signage at the entrance, requiring each visitor to be accompanied by at least one other person, and instituting a ticket price of USD 10 (previously free) [5,6]. However, these interventions did not stop the fourth suicide, which occurred just two months after *Vessel* reopened and six months after the third death [7]. *Vessel* has since closed again, and Hudson Yards is currently exploring and actively evaluating options that would allow it to reopen, including installing safety netting under the flights of stairs [8]. However, as of the time of writing, there is no exact timeline for the reopening.

Social media facilitates the discussion of suicides at suicide hotspots through a connected and borderless network. In the case of *Vessel*, by searching relevant hashtags such as #vessel and #vesselsuicide, people can access a huge amount of discussion on the topic. A high volume of discussions about *Vessel* as a specific, accessible, and public suicide site emerged over the 29-month period in question; these discussions have the potential to reflect public sentiment concerning suicide cases and to urge the implementation of effective measures from relevant responsible authorities. This is particularly relevant regarding suicide prevention at a ‘suicide hotspot’ [9], also referred to as ‘frequently used locations’ and ‘high-risk places’ [10]. In this paper, we adopt the term ‘suicide hotspot’ for precision and brevity.

Moreover, social media posts and their content can be used to assess the extent of influence on users participating in those public discourses [11,12]. They can also provide invaluable insights into public discussions and views on various topics, which may be useful for policymakers when making decisions [13,14,15,16]. Therefore, examining how discussions on social media change over time can help monitor changes in public attitudes concerning suicide prevention.

To our knowledge, no studies have documented the change in frequency and content of social media posts about suicide acts (and the responses to these acts) at a suicide hotspot. Some studies have investigated traditional print media’s coverage of suicide hotspots, and have suggested future studies on online and social media content [17,18]. Here, using a bottom-up approach, we examine the trajectory of suicide-related activity about *Vessel* on Twitter in order to understand the changes in and focus of public sentiment, which may be pivotal in deciding the future of *Vessel*. Furthermore, by identifying these patterns, we offer concrete suggestions to establish more preventive solutions if *Vessel* reopens. 

## 2. Materials and Methods

We investigated public opinion relating to *Vessel* and suicide through relevant tweets extracted from Twitter. We conducted an engagement analysis to explore the trajectories of the discussion, and thematic analyses to examine the nature of the discussion. The volume of and changes in engagement, as well as the themes of tweets during the period of the four suicides, were used as a proxy for understanding the changes and focus of public sentiment related to suicide deaths at *Vessel*. 

### 2.1. Data

We accessed relevant tweets using the Meltwater (Meltwater is a software company that provides both online and printed media data for media monitoring and social listening services. URL: https://www.meltwater.com/en (accessed on 15 September 2022)) database. Specifically, we extracted all tweets (*n* = 30,609) that mentioned “*Vessel*” (case insensitive) from February 2020 to August 2021. To remove irrelevant tweets, we only included those that also mentioned the location of the architecture, including “New York,” “NYC,” and “Hudson.” In total, 9130 tweets met our criteria. We further extracted a subset of tweets that mentioned suicide-related topics using a list of keywords (“suicide,” “die,” “death,” “jump”), and identified 3774 tweets. The keywords were identified from the pool of the top 500 keywords from the 9130 relevant tweets, and then the keywords with suicide-related meanings were extracted by social workers. Finally, we removed tweets posted by organizational accounts (representing any organizations rather than individuals) because they did not represent public opinion. We also removed outliers posted after the first 14 days following each suicide case, resulting in a total of 3058 tweets which were then subject to engagement analysis and thematic analysis. Figure 3 summarizes the datasets included in the engagement and thematic analyses, each of which is described in more detail below. 

### 2.2. Engagement Analysis

Tweets posted by personal accounts that mentioned *Vessel* and suicide during the first 14 days (from the day of the suicide case and for the following 13 days) of each suicide case were extracted for the engagement analysis (*n* = 3058). Engagement usually refers to users’ participation in a public discussion [15]. In our study, we used: (1) the number of tweets (including retweets), and (2) the length of the period during which related tweets continued to be posted and shared on the media platform; together, these represent user engagement in the discussion of the given case. The larger the number of tweets and the longer the period during which related tweets emerged on the platform, the higher the user engagement, or, the more significant the impact of the relevant suicide cases. We used the probability density function (PDF) and cumulative distribution function (CDF) to evaluate and compare the intensity of and the trends in user engagement in the discussion of each of the four suicide cases. The KS statistic of Kolmogorov–Smirnov test was used to compare the PDF and CDF distributions of the discussions of the four suicide cases [19]. 

### 2.3. Thematic Analysis

For the thematic analysis, we also used tweets from personal accounts that mentioned Vessel and suicide in the first 14 days after each suicide case. Because of the imbalanced number of tweets across the four cases, we followed the previously used process of oversampling minorities [20]. We over-sampled the tweets for the first two cases and under-sampled the tweets for the third and fourth cases due to the different volumes of data in the earlier (first and second) and later stages (third and fourth). Specifically, we extracted the complete set of tweets for the first two cases and pulled a random sample of 10% of the tweets for each of the last two cases.

The conventional content analytical approach was used as a framework for the thematic analysis [21]. We developed a tentative coding scheme based on the independent analysis of a random trial subset of 500 tweets using opening coding by two authors (the fourth author, a Masters-level research assistant, and the fifth author, a certified counselor). The coding scheme was refined and inconsistencies were resolved through revisiting the relevant data and having team discussions to reach an agreement. Some themes were combined during the process. The codebook, with examples of each theme, can be found in Appendix A. None of the themes are mutually exclusive and each tweet could be coded into multiple themes based on its content. The sampled tweets (*n* = 506) were independently analyzed using the final coding scheme by the two authors who developed it, and by three additional coders to test their inter-rater reliability (Krippendorff’s α = 83% [22]). The chi-square test was used to examine the differences in the frequency of each theme across the four suicide cases. 

The last theme, ‘suggestions on suicide prevention measures’, was further analyzed and categorized into subthemes using the same coding scheme development process. The codebook, with examples, can be found in Appendix A. 

## 3. Results

Table 1 reports the characteristics and the number of tweets sampled for each suicide case. In total, 3165 *Vessel*-related tweets mentioned suicide keywords, counting from the day of each suicide case for the following 13 days. Tweets posted by personal accounts make up the majority (97%) of the dataset, while tweets from organizational accounts represent just 3%. Tweets from organizational accounts show a downward trend, from 22% for the first suicide case to 3% for the fourth suicide case. Tweets from organizational accounts were removed before being sampled for thematic analysis. In the end, following the sampling strategy stated in the method section, a total of 506 tweets were sampled for thematic analysis. 

### 3.1. Engagement Analysis

Figure 4 shows the trend and distribution plots of the intensity of tweets from individuals mentioning *Vessel* and suicide within the first 14 days of each suicide case (*n* = 3058). It took one day for the number of tweets to peak for the first, second, and fourth cases, and three days for the third case. However, the discussion on Twitter became protracted and was amplified as more suicides were reported. More users joined in the discussion of the incidents, and it took longer for those discussions to fade out. The average daily number of tweets for the first and second cases was six (*SD* = 14) and ten (*SD* = 23), respectively. The third and fourth cases gained more attention, with the average daily number of tweets being 98 (*SD* = 197) for the third case and 104 (*SD* = 129) for the fourth case. 

The CDF plots in Figure 5 show that the cumulative discussion stopped increasing from day four for the first case. In contrast, the cumulative debate for the fourth case lasted longer, with the increase extending the full 14 days (KS statistic = 0.93, *p* < 0.001). The PDF plots also show that it took longer for the discussions of the fourth case to dissipate than those of the first case (KS statistic = 0.71, *p* < 0.001). 

### 3.2. Thematic Analysis

Table 2 reports the percentages of the six main themes based on the 506 sampled suicide-related tweets for the thematic analysis. 

The majority of the tweets sampled over the four cases involved elements of news reporting, ranging from 93.5% to 98.6%. The percentage of tweets involving expressions of emotion and personal opinions about the suicide incidents increased from the first death (21.6%) and peaked with the third death (47.8%), but dropped back to 15.8% with the fourth. The analysis also found a small number of self-focused messages, including people’s complaints about the closure of *Vessel* due to the suicide incident, their experiences at *Vessel*, and their desire to visit it; the chi-square test shows no statistical difference in the percentage of tweets across the four cases. We also identified a trend relating to complaints and criticism about *Vessel* and its developer, which increased across the four deaths, from 1.3% after the first death to 23.3% after the last death; here, a significant difference can be observed. Tweets that included supportive messages were minimal across the four cases, with no significant differences. Regarding suggestions on suicide preventive measures, significantly more Twitter users commented on this theme after the third and fourth deaths in comparison to the first two cases. 

The tweets from Theme 6, ‘suggestions for suicide preventative measures’, were further categorized into smaller subthemes. Table 3 reports the percentages of the seven types of suggested solutions over a total of 506 sampled suicide-related tweets. 

Following the first and second cases, there were minimal tweets suggesting suicide preventative measures, and they were mostly general or vague suggestions. The majority of suicide prevention suggestions were found following the third case, where many people emphasized the need to improve access to mental healthcare (33.3%). However, this percentage dropped to 1.7% in the fourth case, and public discussion shifted towards suggesting restricting means, including closing the site permanently (3.4%), dismantling the sculpture (14.4%), as well as erecting barriers (9.6%). 

## 4. Discussion

Using the social media platform Twitter, this research documented the changes in public attitudes towards suicides at *Vessel* since the first incident occurred; we used two approaches. Our engagement analysis shows a cumulative and lingering effect of suicide-related discussion after the second suicide. Engagement significantly increased for the third and fourth suicide cases, and the discussion continued for longer and took more time to dissipate. The thematic analysis shows that most tweets from personal accounts are either retweets from news accounts or mention news headlines about the suicide cases; moreover, many of these discussions are characterized by emotional responses, suggesting that online media reporting on suicides at *Vessel* had a cumulative effect on the public mood. However, it is interesting to note that the number of tweets expressing emotions and personal opinions on the suicide incidents dropped for the fourth suicide after a steady increase from the first to the third one. Additionally, since the fourth suicide, the tone of the discussions has shifted toward demanding both that people be held accountable and that more effective suicide prevention strategies be implemented. 

It is worth noting here that, like reporting in traditional media, discussions on social media may not always have a positive impact [23]. Even well-intentioned messages designed to raise awareness or galvanize community action may attract attention to a suicide hotspot, drawing vulnerable individuals to the site and increasing the likelihood of suicide contagion [9,23]. Meanwhile, research has shown that high levels of exposure to media coverage of traumatic events is associated with more severe symptoms of post-traumatic stress disorder [24,25]. Excessive retweets and discussions of the incidents may prolong or aggravate the negative impact on witnesses and suicide survivors [26,27,28]. Consideration should be given to how social media might complement other interventions to prevent suicides and the collateral damage at suicide hotspots. In the traditional media arena, guidelines designed to encourage responsible news reporting may help to ensure that newspaper, television, and radio reports are measured [29]. In social media, novel interventions are needed to steer discussions in positive directions to prevent unintended consequences at suicide hotspots, especially when the volume of Twitter discussions showed an upward trend after each successive suicide case in this study. While some have pointed out the difficulty in establishing detailed legal rules on social media platforms and emphasized their importance in providing a public space for individuals to grieve over the loss of lives and to show empathy [30,31], our human annotation indicates that the number of supportive messages, including those that encourage help-seeking, were minimal across the four cases. Thus, relevant guidelines should be publicized and the public should be educated; #chatsafe is one such intervention that is gaining traction [32,33,34]. However, it has not yet been used in the context of suicide hotspots. If public adherence to these guidelines is deemed difficult, better dissemination strategies and closer monitoring of tweets posted by organizational accounts, especially news accounts or other verified accounts with a larger volume of followers, are necessary [35]. 

Suicide is a public health problem [36]. Our results also show that many people recognized the importance of improving access to mental healthcare following the third case. However, when it came to the fourth suicide, the percentage dropped dramatically, and the majority of those who suggested suicide preventative measures called attention to the pressing need to implement effective, structural interventions at the site to prevent further suicide incidents. The literature identifies three main approaches to suicide prevention that are sometimes taken at suicide hotspots, and several systematic reviews and meta-analyses have evaluated and compared their effectiveness [37,38,39,40]. The evidence for restricting access to means was particularly strong, whereas evidence for encouraging help-seeking and increasing the likelihood of intervention by a third party (e.g., adding security measures) was relatively weak. The results of this study indicate that the fourth case was the tipping point for the community to make specific suggestions regarding restricting access to means, including raising barriers, dismantling the structure, and permanent closure of the site. This trend is consistent with the idea of modifying the structure of Vessel, which was discussed in several architectural periodicals soon after the fourth death [41,42,43]. The suggestion of dismantling *Vessel* and recycling its parts for other uses, such as pedestrian bridges and viewing platforms, was also put forward [43]. 

Although some steps were taken to prevent suicide at *Vessel* after the second and third suicides occurred, these would be regarded as “light touch” interventions at best. This opinion is also reflected in the sampled tweets in our analysis and in the evidence from systematic reviews and meta-analyses [37,38]. The site was initially temporarily closed, and when it reopened, visitors had to enter *Vessel* in pairs and were charged a fee. The former was presumably designed as a measure to increase the likelihood that a third party would be in a position to intervene if someone did appear to be starting to take life-threatening actions. The latter was presumably designed to limit access to the site and increase the likelihood that people would be entering *Vessel* for its intended purposes as a visitor attraction. Unfortunately, these measures were inadequate to stop the fourth suicide and possible future ones.

As noted previously, the most effective intervention at suicide hotspots is restricting access to means; this has been shown across several studies to reduce suicides by around 90%, and usually without substitution effects [37,38,40,44,45,46,47,48]. In most cases, restricting access to means involves erecting high barriers that prevent jumping. This intervention was implemented in an upscale 13-story shopping mall in Hong Kong after three suicides by jumping in 2004 [49]. After the erection of a high ring fence (about 1.7 m, rather than the standard 1.1 m), there were no further suicide attempts over the following decade. Similarly, NYU installed floor-to-ceiling perforated aluminum screens as a prevention measure in its 12-story Bobst Library after three student suicides, and no further suicides have been reported since [50]. 

Despite strong and substantial evidence showing the effectiveness of means restriction in preventing suicide by jumping at suicide hotspots, there is often resistance from the community, policymakers, or the owners and developers of the site to taking any proactive structural actions, such as erecting barriers [38,39]. For bridges, one of the key arguments against the installation of barriers is its high cost; for cliffs, it is usually because it would mar the natural scenery; and for buildings, it is typically due to costs and the resistance to altering the buildings’ appearances. In our sampled tweets, although a few people complained about the closure of *Vessel* due to the suicide incidents and expressed their wish to visit it in the future (Theme 3)*,* no one argued against the installation of barriers for aesthetic reasons; in fact, this was the second most popular suggestion, closely following the demand for dismantling the structure in the fourth case. Regarding the economic argument surrounding raising barriers, although the price is indisputably high, studies have shown that the return on investment often outweighs its cost [51,52]. Erecting barriers is proven to be a highly cost-effective and warranted suicide prevention strategy that is well recognized by the public. 

Unfortunately, delays in installing safety barriers and fences have failed to save lives at suicide hotspots [47,52,53,54,55]. For the suicide hotspots studied in a meta-analysis by Pirkis et al., it took at least five suicide deaths before some interventions were implemented [37]. The findings of our study demonstrate that public discussions on social media mirror the typical pattern for demanding effective and specific interventions at high-risk locations. The thematic analysis shows that the emotional content of tweets grew from the second case but then shifted to concerns about how to implement effective preventative measures following the third and fourth deaths. The social media activity reflects changing community attitudes, such as people starting petitions against *Vessel*. This demonstrates increasing demand and pressure from the general public for serious actions to prevent additional incidents. 

As of writing, *Vessel* remains closed and is currently exploring and evaluating options that would allow it to reopen [8]. We suggest that *Vessel* should leverage Twitter data as a proxy for public opinion and research evidence before making any decisions. If it is to reopen, it would at the very least require effective, higher barriers to be installed; this is the second most popular suggestion, following dismantling the building. There is a question as to whether doing this would be possible, and even if it was, whether it would be sufficient. The nature of the site means that part of its attraction is the view and its creative design, which raised glass barriers may tarnish. There are multiple other examples of suicide hotspots where, for example, the site’s beauty has been taken into account when safety barriers were erected. Still, most of these sites have been bridges or cliffs rather than artistic installations. There would undoubtedly be complex aesthetic and engineering considerations associated with raising the barriers at *Vessel*. However, we are confident that the creativity of its architect is sufficient to identify an architectural solution that considers both its form and its function in keeping visitors safe. If doing so proves too difficult, we suggest that *Vessel* should be permanently closed. 

### Limitations

First, since we sampled only 10% of the tweets from the third and fourth cases in the thematic analysis, there may be a sampling bias whereby some discussions were ignored. The volumes of tweets in the last two cases may also have been disproportionately downplayed. Second, there was no demographic information behind the tweets, as all personal data linked to the accounts were removed due to Meltwater’s agreement with Twitter. Third, we did not include other engagement indicators, such as sentiment analysis, to categorize the tweets into positive, negative, and neutral sentiments; the main reason for this decision is that this one-dimensional classification might be too simplistic to reflect the actual situation (e.g., it is common to find multiple sentiment-related words in one tweet—for example, both expressing grief and showing support to others—which would be difficult to classify into one single sentiment). Thirdly, we did not account for the responses and interactions of the sampled tweets (e.g., the number of replies, likes, and followers of the user). Lastly, we only included data from Twitter. However, there were substantial discussions of *Vessel* suicides on other platforms (e.g., Reddit and print media). Future studies should focus on using a larger, more representative sample, examining the public sentiment by utilizing additional engagement indicators on various social media platforms, as well as their adherence to #chatsafe guidelines. 

## 5. Conclusions

Our analyses of Twitter data after the four individual suicide cases at *Vessel* demonstrated that the negative emotional content of discussions grew after the second suicide case, but shifted to concerns about how to implement structural preventative measures following the fourth case. The public sentiment evolves dynamically with the increasing number of cases, and is consistent with the typical pattern for demanding specific structural interventions at high-risk locations, as identified in the scientific literature as well as in the architectural field. So far, the responses of *Vessel* have not met public expectations in mitigating the suicide risk. Restriction of means is one of the most effective evidence-based suicide prevention methods. If *Vessel* is to reopen, structural modifications should be implemented to avert further suicides. We understand that this could create significant legal, logistical, and financial issues, and therefore suggest that different sectors (financial, legal, public relations, government etc.) should work together with architects and suicide experts to find a solution, and to build a safe place for visitors while maintaining the boundaries for creativity. If proven too difficult, then *Vessel* should perhaps remain permanently closed for the sake of protecting the vulnerable population, as suggested by the community. Our results show that the public recognizes the importance of taking the potential risks of suicide into account when designing architecture. We hope the restriction of means can be implemented earlier to prevent further loss of life, especially when we have reason to believe—based on prior experience elsewhere—that it will happen again. We should not wait for the fifth death before implementing effective preventive actions. 

## Figures and Tables

**Figure 1 ijerph-19-11694-f001:**
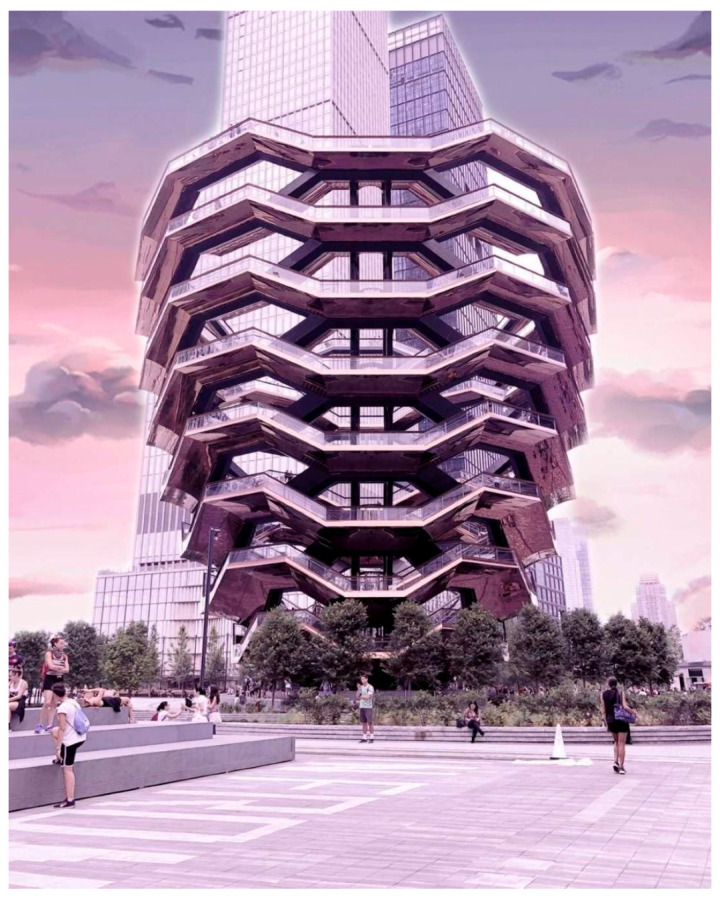
*Vessel* (Photo credit: Yunyu Xiao).

**Figure 2 ijerph-19-11694-f002:**
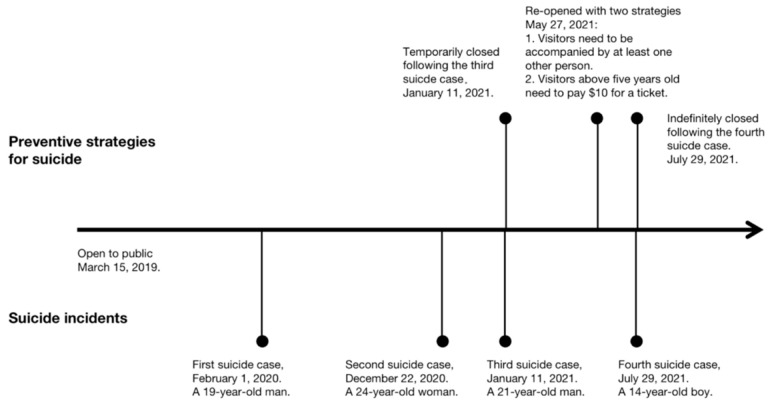
Timeline of suicide incidents at *Vessel* and the preventative strategies that followed (as of August 2021).

**Figure 3 ijerph-19-11694-f003:**
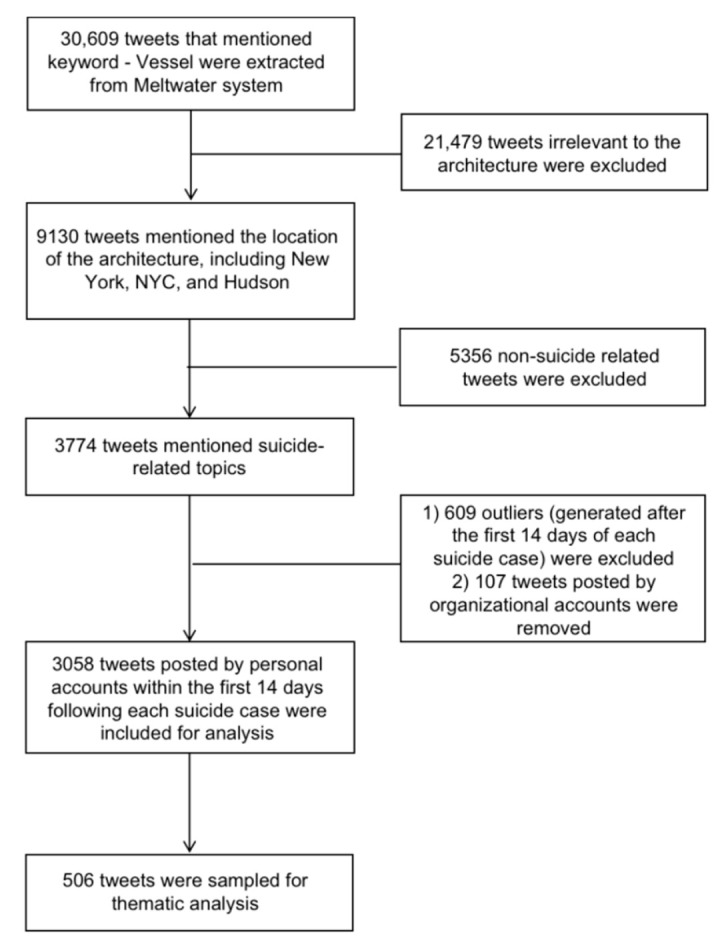
Dataset summaries and criteria for inclusion in engagement and thematic analysis, from February 2020 to August 2021.

**Figure 4 ijerph-19-11694-f004:**
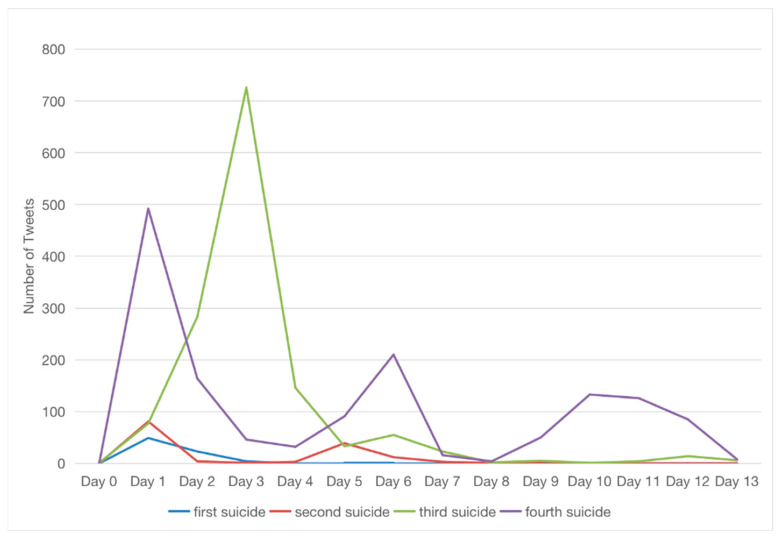
The number of tweets mentioning *Vessel* and suicide within the first 14 days after each suicide case.

**Figure 5 ijerph-19-11694-f005:**
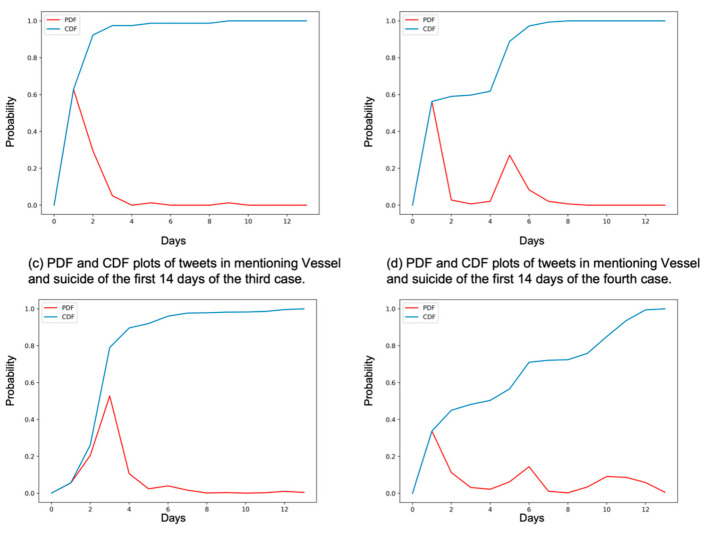
(**a**–**d**) are the probability density function (PDF-red) and cumulative distribution function (CDF-blue) plots of tweets mentioning *Vessel* and suicide within the first 14 days of the first, second, third, and fourth suicide cases, respectively.

**Table 1 ijerph-19-11694-t001:** Characteristics of tweets and the number of tweets sampled for thematic analysis.

Suicide Case	Total Tweet Count	Tweets Posted by Personal Accounts	Tweets Posted by Organizational Accounts	Number of Tweets Sampled for Thematic Analysis
First case	100	78 (78%)	22 (22%)	78
Second case	154	144 (94%)	10 (6%)	144
Third case	1409	1376 (98%)	33 (2%)	138
Fourth case	1509	1457 (97%)	45 (3%)	146
Total	3165	3058 (97%)	107 (3%)	506

**Table 2 ijerph-19-11694-t002:** Summaries of the distribution of themes in tweets across the four cases.

Themes	First Case (%) (*n* = 78)	Second Case (%) (*n* = 144)	Third Case (%) (*n* = 138)	Fourth Case (%) (*n* = 146)	χ^2^
1—News reporting	96.2_a_	97.2_a_	93.5_a_	98.6_a_	5.85
2—Expressions of emotion or personal opinions	21.8_a_	24.0_a_	47.8_b_	15.8_a_	40.70 ***
3—Self-focused messages	1.3_a_	1.4_a_	5.8_a_	3.4_a_	5.54
4—Complaint or criticism about the building/architect/developer	1.3_a_	4.2_a_	5.8_a_	23.3_b_	44.58 ***
5—Supportive messages	2.6_a_	7.6_a_	1.4_a_	2.1_a_	9.96 *
6—Suggestions of suicide preventative measures	1.3_a_	4.2_a_	34.8_b_	23.3_b_	63.62 ***

Footnote. Chi-squared tests and post-hoc pairwise comparisons were performed with Tukey’s HSD adjustment. Subscripts a and b are used to represent the difference between groups. Groups without statistical differences share the same subscript. Groups that do not share the same subscripts differ by * *p* < 0.05, *** *p* < 0.001.

**Table 3 ijerph-19-11694-t003:** Summaries of the distribution of suggestions of suicide preventative measures.

Themes	First Case (%) (*n* = 78)	Second Case (%) (*n* = 144)	Third Case (%) (*n* = 138)	Fourth Case (%) (*n* = 146)
1—Add netting	1.2	0.0	0.0	0.0
2—Erect barriers	0.0	0.0	0.0	9.6
3—Permanent closure of the site	0.0	0.0	0.7	3.4
4—Dismantle the sculpture	0.0	0.0	0.0	14.4
5—Improve access to mental healthcare	0.0	0.0	33.3	1.7
6—Adhere to media guidelines	0.0	2.1	0.0	0.0
7—General/vague/other preventative measures	0.0	2.8	0.0	0.14

## Data Availability

All data were extracted from Meltwater, a software company that has obtained licenses from Twitter and other online and print media. Please refer to https://www.meltwater.com/en for more details or inquiries.

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
