# Peer review of "Social Media Sentiments on Suicides at the New York City Landmark, Vessel: A Twitter Study"

_ijerph, 2022, doi:10.3390/ijerph191811694_

Round 1
Reviewer 1 Report
1. This paper studied the trajectory of public sentiment on suicide-related activity at Vessel on Twitter by investigating the engagement patterns and identifying themes about the four suicides from 2020-2021. It yields interesting findings and provides insights into public sentiment on suicide-related activity.
2. The abstract is clearly written. It provides an appropriate statement of study results and how it was achieved.
3. The introduction and the literature review were adequate.
4. There are 6 self-citations by one of the authors, Jane Pirkis. Although the self-citations seem relevant to this paper, 6 self-citations might be too many.
5. The conclusion is rather short considering this is a very complex and sensitive topic. In addition, suggestions for further study should be provided especially if the limitations are listed.
Reviewer 2 Report
Important study, well done and well-written. Would have been nice to know more about the demographics behind the tweets. The biggest challenge that I have with the study are the conclusions that the Vessel should remain closed if appropriate barriers are not created. I think that is a dangerous place the authors go to in terms of determining what public businesses, structures, etc. must have. If accepted as is, this could have monumental problematic implications for buildings, bridges, and all structures currently in place that are hot spots, places that potentially could become hot spots and future designs. While this might be a good thing, it is unrealistic and potentially creates legal and financial issues, as well as logistical issues around the world. Thus I strongly suggest re-working this conclusion.
